# The Consequences of the Pandemic on Medical Students’ Depressive Symptoms and Perceived Stress: A Repeated Cross-Sectional Survey with a Nested Longitudinal Subsample

**DOI:** 10.3390/jcm11195896

**Published:** 2022-10-06

**Authors:** Giuseppina Lo Moro, Sara Carletto, Vittoria Zuccaroli Lavista, Giovanna Soro, Fabrizio Bert, Roberta Siliquini, Paolo Leombruni

**Affiliations:** 1Department of Public Health Sciences, University of Torino, 10126 Torino, Italy; 2Department of Neuroscience “Rita Levi Montalcini”, University of Torino, 10126 Torino, Italy; 3Clinical Psychology Unit, A.O.U. City of Health and Science of Torino, 10126 Torino, Italy; 4Formerly at School of Medicine, University of Torino, 10126 Torino, Italy; 5Hospital Medical Direction, A.O.U. City of Health and Science of Torino, 10126 Torino, Italy

**Keywords:** COVID-19, medical students, depression, stress, prospective study

## Abstract

This study aimed to explore the impact of the pandemic on medical students’ mental health in Italy using a repeated cross-sectional survey with a nested longitudinal subsample (first timepoint: 2018; second: 2020/2021). Three research questions (RQs) were investigated. Study 1 (longitudinal sub-sample) explored whether medical students had higher levels of depressive symptoms and stress during the pandemic compared with a pre-pandemic period (RQ1) and what variables were associated with these conditions during the pandemic adjusting for baseline levels (RQ2). Study 2 (repeated cross-sectional data) aimed to examine whether medical students had higher levels of these conditions during the pandemic compared with their same-year peers during a pre-pandemic period (RQ3). In Study 1, higher levels of depressive symptoms and stress were shown during the pandemic (RQ1). Multivariable models highlighted associations between poor mental health and worsening of the judgment of medical school choice, worsened psychological condition due to the pandemic, economic repercussions due to the pandemic, and baseline levels of symptoms (RQ2). In Study 2, our findings reported higher levels of depressive symptoms and stress during the pandemic, also adjusting for other variables (RQ3). In conclusion, depressive symptoms and stress were greater during the pandemic. The most relevant variables were pandemic-related items and medical school choice judgment.

## 1. Introduction

Medical students have been recognized as a population at risk for poor mental health outcomes [1,2,3,4,5,6,7]. In particular, medical students may have a high risk of depression [1,2,3], with a pooled prevalence of depressive symptoms of 27.0% (95% confidence interval (CI), 24.7–29.5%) [1], and a high levels of psychological distress [4,5,6], with some data suggesting that about half of medical students might suffer from burnout during their medical education [7]. In addition to a high risk of mental issues, medical students have been reported to have poor help-seeking behavior and undergo treatments infrequently [2], mainly because they have the perception that having mental disorders is a weakness and hinders their medical career [8]. Moreover, severe psychological distress is not only a problem per se but it can be associated with poor academic performance and serious health consequences, such as substance use and suicidal behaviors [9]. Last, poor mental health can persist when medical students become residents and can lead to care of low quality and a higher likelihood of medical errors [10].

In the context of the current pandemic, which is leading to several mental health consequences both among healthcare workers and among the general population [11,12,13], it could be possible that medical students have also experienced a mental health deterioration. They may have gone through concerns commonly shared among youth, e.g., safety issues, fear of transmission of disease, massive e-learning adoption, and limited social interactions [13], and through concerns specifically related to their education, e.g., suspension of clinical training, alterations in grading, changes in medical licensing exams, and perceived lack of preparation for the application to residency programs [14,15,16]. Indeed, available reviews about mental health among medical students during the pandemic have highlighted high levels of depression, anxiety, stress, and exhaustion, suggesting a rise in those symptoms [17,18]. However, most of studies had a cross-sectional [17] or a longitudinal design [19,20,21] without taking into account pre-pandemic periods, thus not allowing inferring about an actual worsening of mental health conditions compared with pre-pandemic years. To date, very few studies focusing on medical students have presented prospective (repeated cross-sectional or longitudinal) data beginning from a pre-pandemic period [22,23,24,25]. Interestingly, such studies found conflicting findings, ranging from a significant increase in stress with unchanged depression [24] to non-significant changes in mental health outcomes [22,25] and even decreased symptoms [23]. Thus, current data are limited and describe areas that may have different epidemiological situations of Coronavirus Disease of 2019 (COVID-19) and different sociocultural backgrounds. 

In particular, Italy was one of the earliest European countries to be severely affected by the pandemic and the first European country to enter a nationwide lockdown in March 2020, with a series of restrictive measures that also involved the closure of universities [26]. Since, to our knowledge, no prospective study on Italian medical students has been performed yet, we aimed to explore the potential impact of the pandemic on medical students’ mental health in the Italian context using a two-timepoint repeated cross-sectional survey with a nested longitudinal subsample. The first timepoint was before the pandemic and the second timepoint was during the pandemic. Specifically, the present paper had three main research questions (RQs). 

The first two RQs referred to the longitudinal subsample:

RQ1: Did medical students have higher levels of depressive symptoms and perceived stress during the pandemic compared with a pre-pandemic period? 

RQ2: What variables were associated with depressive symptoms and perceived stress during the pandemic adjusting for the baseline levels before the pandemic)?

The last RQ considered the repeated cross-sectional survey data. Indeed, we hypothesized that some changes in mental health in the longitudinal subsample might not be due to the pandemic but to other factors, including the transition from the first year of medical school to a later one. Thus, for the third RQ we took into account medical students who attended the same years of medical school during the first and the second timepoint:

RQ3: Did medical students have higher levels of depressive symptoms and perceived stress during the pandemic compared with medical students who attended the same year of medical school during a pre-pandemic period?

## 2. Materials and Methods

### 2.1. Study Design and Sample

The present study was a two-timepoint repeated cross-sectional survey with a nested longitudinal subsample. The first timepoint was before the pandemic and the second timepoint was during the pandemic. Considering both the timepoints, all procedures performed were in accordance with the 1964 Helsinki declaration and its later amendments. The Ethics Committee of the University of Turin approved the two protocols. Participants were asked to sign an informed consent form. Participation was voluntary and anonymous, and participants received no compensation.

The study conducted at the first timepoint was performed around November 2018: a multicenter cross-sectional survey, called Psychosocial Report in Italian MEdical Students (PRIMES), was carried out in 12 Italian medical schools, including the medical school at the University of Turin [27,28,29,30]. The sub-sample from the University of Turin consisted of 506 participants. Medical students were enrolled in the classrooms by convenience in the 1st, 4th, and 6th years of their programs. Participants of the 1st year attending the University of Turin were asked to create a nickname according to rules decided by the researchers to potentially match the data in later studies. The main aims of PRIMES were to estimate the levels of depressive symptoms and perceived stress among Italian medical students and explore factors associated with these two conditions. The first timepoint results have been fully described elsewhere [27,28,29,30]. 

The study executed at the second timepoint was conducted between December 2020 and February 2021: a cross-sectional survey was carried out at the medical school of the University of Turin. Medical students were recruited by convenience through emails from the 2nd to the 6th years of their programs (1st year students were excluded since they had just begun the academic year and did not live the pandemic as medical students). The final sample consisted of 1329 participants. To contextualize, in Italy the universities were closed in March 2020 (national lockdown from March to May 2020) and online lectures and exams were carried out until June 2021. The questionnaire was developed based on the PRIMES questionnaire. Students were asked to create a nickname according to the same rules as in PRIMES; therefore, it was possible to match 121 unique participants (attending the 1st year at the first timepoint and the 3rd year at the second timepoint). The full results of the cross-sectional survey conducted at the second timepoint are presented in another paper [31]. 

In the present paper we merged the first timepoint (only considering data from a single center, i.e., University of Turin) and the second timepoint data. To answer the three RQs, we decided to create two sub-studies that had two different datasets:

(a). Study 1: Longitudinal subsample: composed by the 121 students whose nickname was identified at both the timepoints (for RQ1 and RQ2);

(b). Study 2: Repeated cross-sectional data: composed by students who attended the 4th and the 6th year at the first timepoint and students who attended the 4th and the 6th year at the second timepoint (a total of 705 students). The other years were excluded since they were different between the two timepoints (for RQ3).

### 2.2. The Questionnaire

The questionnaires at the two timepoints had a core set of items that were identical, based on the PRIMES study. The researchers of PRIMES developed the survey after exploring the available literature on depression and stress among medical students. Details on the literature on which the variables were based on are provided in the PRIMES paper on depressive symptoms [27]. Specifically, these core items referred to age, gender, sexual orientation, relationship status, living condition, being an off-site student, family cohesion, economic situation, family history of psychiatric disorders, judgment about school choice, friendships with classmates, and climate among classmates. This set of variables was selected to be included at the second timepoint because such variables were associated with depressive symptoms [27] or perceived stress [28] at the first timepoint. In addition, variables describing changes in the above-mentioned items were calculated and used in Study 1 (RQ2). These variables are fully described in the Appendix A.

Then, at both timepoints, the Beck Depression Inventory-II (BDI-II) and the Perceived Stress Scale-10 (PSS-10) were used. The BDI-II is a 21-item self-report instrument that evaluates the severity of depressive symptoms [32]. The total score ranges from 0 to 63, where higher scores represent a higher severity. BDI-II internal consistency has been reported to be around 0.9 [33]. In our sample, Cronbach’s Alpha was above 0.91 at each timepoint in both Study 1 and Study 2 (except for the first timepoint in Study 1: 0.89). To describe the presence of depressive symptoms, a cut-off of 14 was used [1]. The PSS-10 is a self-reported tool to assess psychological distress [34]. The total score ranges from 0 to 40, where higher scores represent higher probability of perceived stress. PSS Cronbach’s alpha reliability coefficient has been calculated to be 0.89 among college students [35]. In our sample, Cronbach’s Alpha was above 0.87 at each timepoint in both Study 1 and Study 2. Scores from 0 to 13 can be considered low stress, scores from 14 to 26 moderate stress, and scores from 27 to 40 high stress [36].

In PRIMES, there are additional items mainly on hobbies/extracurricular activities, career motivations, grade average, and exams [27]. This set of variables was not used in the present paper.

At the second timepoint, a Patient and Public Involvement (PPI) framework was used to change and update PRIMES items. Indeed, there were additional questions mainly on fear of contagion, impact of the pandemic on lifestyles and academic life, feelings of loneliness, economic repercussions, existential reflections, and overall psychological condition due to COVID-19. This set of variables was not used in the present paper, except for the variables selected in the final models of the cross-sectional survey [31] as explained below in Study 1 (RQ2). These variables are fully described in the Appendix A.

### 2.3. Statistical Analysis

Descriptive analyses were performed for all the variables. Categorical variables were described as frequencies and percentages. Continuous variables were expressed as median and interquartile range (IQR) as they did not show a normal distribution (Shapiro Wilk Test); except for the PSS-10 score in Study 1, which reported a normal distribution and, therefore, it was described as mean and standard deviation (SD).

Overall, the analyses were performed using IBM SPSS Statistics software (IBM Corp. Released 2020. IBM SPSS Statistics for Windows, Version 27.0. IBM Corp, Armonk, NY, USA). Stata 16 (StataCorp. 2019. Stata Statistical Software: Release 16. StataCorp LLC, College Station, TX, USA) was used to run the repeated measures multilevel models in Study 1. A two-tailed *p*-value < 0.05 was considered to be significant. Missing values were excluded by pairwise deletion in descriptive analyses and by listwise deletion in regressions.

#### 2.3.1. Study 1: Longitudinal Subsample

Mann Whitney U test (for BDI-II) and *t*-test (for PSS-10) were computed to test whether the participants whose nickname was not matched had a different level of symptoms compared with the selected subsample.

The McNemar test was used to assess differences in the set of core variables identical at the two timepoints.

Mann Whitney U test (Kruskal Wallis test where appropriate) was used to study the distribution of BDI-II across the independent variables, including the variables created to describe change and the variables recorded only at the second timepoint. With the same purposes, *t*-test (one-way analysis of variance ANOVA where appropriate) was used for PSS-10. Spearman’s rho was used for age.

To answer RQ1, the related-samples Wilcoxon signed rank test and the paired-samples *t*-test were executed to compare the BDI-II and PSS-score between the timepoints. Moreover, the McNemar test and the Marginal Homogeneity test were used to compare the presence of depressive symptoms and the categories of stress.

Moreover, to answer RQ1 adjusting for relevant variables, the scores of BDI-II and PSS-10 considering both timepoints (long data format) were the outcomes of regression models. For each outcome, a repeated-measures multilevel mixed-effects linear regression model was executed (levels: participant, timepoint). In addition to the timepoint, the model was adjusted for age, gender and the independent variables that were available both at the first and at the second timepoint and were associated with the outcome in at least one of the two timepoints. All the independent variables were time-varying, except gender. Results were expressed as unstandardized coefficients (B) and 95% Confidence Intervals (95% CI). 

To answer RQ2, the outcomes consisted of the scores of BDI-II and PSS-10 at the second timepoint (wide data format). For each outcome, a multivariable linear regression model (adjusted for age, gender, and the outcome score at the first timepoint) was carried out. The independent variables were selected via a backward stepwise method. In the first step the variables describing changes and the variables selected in the models of the second timepoint cross-sectional study were entered. Results were expressed as unstandardized coefficients (B) and 95% Confidence Intervals (95% CI).

#### 2.3.2. Study 2: Repeated Cross-Sectional Survey

To answer RQ3, Mann Whitney U tests were used to assess the changes in BDI-II and PSS-10 score distributions between the timepoints. 

Moreover, chi-squared tests (Mann Whitney U test for age) were used to assess differences between the first and the second timepoint regarding the independent variables and the categories defined by the cut-off of BDI-II and PSS-10 above-described [1,36]. 

Mann Whitney U tests were conducted to test the distribution of BDI-II and PSS-10 scores across the independent variables. Spearman’s rho was calculated to evaluate the correlations between age and the outcomes.

To study if a potential change in the scores was significant also adjusting for variables that might influence the outcomes, a multivariable linear regression model (adjusted for age, gender, timepoint) was executed for each outcome (BDI-II and PSS-10 scores). The independent variables that were available both at the first and at the second timepoint and were associated with the outcome in at least one of the two timepoints were entered in the model. Results were expressed as unstandardized coefficients (B) and 95% Confidence Intervals (95% CI).

## 3. Results

### 3.1. Study 1: Longitudinal Subsample

The sample consisted of 121 participants. At the first timepoint, first year participants were 222. The distribution of BDI-II score (*p* = 0.904) and PSS-10 score (*p* = 0.681) was not different between the 121 students whose nickname was matched and the 101 students whose nickname was not matched. At the second timepoint, third year participants were 217. The distribution of BDI-II score (*p* = 0.430) and PSS-10 score (*p* = 0.465) was not different between the 121 students whose nickname was matched and the 96 students whose nickname was not matched. Among the 121 participants, 115 students (95.0%) and 116 students (95.9%) completed the BDI-II and the PSS-10, respectively, both at the first and second timepoint.

Females accounted for 68.6% and the median age at the first timepoint was 19 (IQR = 19–20). Between the two timepoints, most of the characteristics of the sample did not significantly change. A significant change was reported only for the judgment of medical school choice (negative judgment from 10.0% to 25.0%), friendships with classmates (unsatisfying friendships from 0.9% to 7.8%), and climate among classmates (competitive and hostile climate from 2.5% to 19.2%) (Table 1). 

The median BDI-II score was 7 (IQR = 3–12) at the first timepoint and 14 (IQR = 7–22) at the second (*p* < 0.001). The mean difference between the BDI-II scores at the second and at the first timepoint was 7 (SD = 9.5). According to the BDI-II cut-off of 14, the presence of depressive symptoms was reported by 20.9% of first timepoint participants and by 50.4% of second timepoint participants (*p* < 0.001) (Table 1). Among students without symptoms at the first timepoint (*n* = 91), 40.7% showed symptoms at the second timepoint. Among students with symptoms at the first timepoint (*n* = 24), 87.5% showed symptoms at the second timepoint. 

The mean PSS-10 score was 16 (SD = 7.2) at the first timepoint and 22 (SD = 8.0) at the second (*p* < 0.001). The mean difference between the PSS-10 scores at the second and at the first timepoint was 13 (SD = 8.2). According to the PSS-10 risk categories, at the first and at the second timepoint, respectively, low risk was reported by 39.7% and by 13.8%, medium risk by 52.6% and 56.9%, high risk by 7.8% and by 29.3% (*p* < 0.001) (Table 1). Among students with a low risk at the first timepoint (*n* = 46), 63.0% and 13.0% reported medium and high risk at the second timepoint, respectively. Among students with a medium risk at the first timepoint (*n* = 61), 8.2% and 32.8% reported low and high risk at the second timepoint, respectively. Among students with a high risk at the first timepoint (*n* = 9), no students reported low risk and one student (11.1%) reported medium risk at the second timepoint.

Considering the BDI-II score, its distribution was significantly different across the categories of sexual orientation (*p* = 0.020), family history of psychiatric disorders (*p* = 0.002), and climate among classmates (*p* = 0.017) during the first timepoint. During the second timepoint, only the relationship with climate among classmates was confirmed (*p* = 0.008) and, in addition, the BDI-II score was differently distributed across the categories defined by family cohesion (*p* = 0.016) and judgment about the choice of medical school (*p* < 0.001). Regarding the PSS-10 score, during the first timepoint it showed a significant relationship only with family history of psychiatric disorders (*p* < 0.001). During the second timepoint, the mean PSS-10 score was different between the categories defined by family cohesion (*p* = 0.001), judgment about the choice of medical school (*p* < 0.001), and climate among classmates (*p* = 0.011) (Appendix A). No significant correlation was found between age and BDI-II score, neither at the first timepoint (*p* = 0.599) nor at the second (*p* = 0.868). Similarly, no significant correlation was found between age and PSS-10 score, neither at the first timepoint (*p* = 0.313) nor at the second (*p* = 0.935).

The variables describing the changes and the distribution of the outcomes across such variables are presented in Appendix A. The variable that reported the greatest change was the judgment of medical school choice: for 75.0% it was unchanged, for 20.0% it worsened, and for 5.0% it improved. The distribution of the BDI-II score was different across the groups defined by the change in judgment of medical school choice (*p* < 0.001) and by the change in judgment of climate among classmates (*p* = 0.043). Higher scores were reported by students who had a worsening in those judgments. The distribution of the PSS-10 score was different across the groups defined by the change in judgment of medical school choice (*p* < 0.001) and by the change in family cohesion (*p* = 0.004). Higher scores were reported by students who had a worsening in those variables (Appendix A). 

The independent variables investigated only at the second timepoint are presented in Appendix A (descriptive analysis and distribution of BDI-II and PSS-10 scores). Because of the pandemic, a total of 62.7% had a feeling of loneliness greater than usual, 39.8% had deeply distressing existential reflections, 55.9% perceived a worsened overall psychological condition, and 17.0% had economic repercussions.

The multivariable regression models to answer RQ1 are presented in Table 2. Completing the survey at the second timepoint was positively associated with the PSS-10 score (*p* = 0.021), but not with the BDI-II score (*p* = 0.059). Having a poor family cohesion was positively associated with both the BDI-II and PSS-10 scores (*p* = 0.001 and p = 0.039), as well as judging medical school choice negatively (*p* < 0.001 and *p* < 0.001). Belonging to a sexual minority had a positive association with the BDI-II score (*p* = 0.006). Having a family history of psychiatric disorders had a positive association with the PSS score (*p* = 0.033).

The multivariable regression models to answer RQ2 are presented in Table 3. The scores of BDI-II (*p* < 0.001) and PSS-10 (*p* < 0.001) during the first timepoint were positively associated with the scores during the second timepoint. A worsened overall psychological condition due to the COVID-19 pandemic was positively associated with both BDI-II (*p* = 0.001) and PSS-10 (*p* = 0.017), as well as a worsened judgment about medical school choice (BDI-II: *p* < 0.001; PSS-10: *p* < 0.001). A poor economic situation had a positive relationship with PSS-10 score (*p* = 0.017), while an improved judgment about medical school choice had a negative relationship (*p* = 0.003).

### 3.2. Study 2: Repeated Cross-Sectional Survey

The sample consisted of 705 participants, 283 (40.1%) from the first timepoint and 422 (59.9%) from the second. Females accounted for 67.2% and the median age was 24 (IQR = 22–24). Sixth year students were slightly more than the half (51.2%). The independent variables had no significantly different distributions across the timepoints, except for age, living condition, and sexual orientation. Descriptive analyses and chi-squared analyses by timepoint are presented in Table 4. 

The median BDI-II score was 12 (IQR = 7–19). It was differently distributed across the timepoints (*p* = 0.001), with a median score of 10 (IQR = 6–18) in the first timepoint and 13 (IQR = 7–21) in the second. Considering the BDI-II cut-off for the presence of depressive symptoms, 42.3% of the sample had depressive symptoms, with a significant difference between the timepoints (*p* = 0.005). Indeed, the presence of symptoms was reported by 35.9% at the first timepoint and 46.6% at the second.

The median PSS-10 score was 21 (IQR = 15–27). It was differently distributed across the timepoints (*p* = 0.002), with a median score of 20 (IQR = 14–26) in the first timepoint and 23 (IQR = 16–27) in the second. Considering the PSS-10 risk categories, 26.2% reported a high-risk, with a significant difference between the timepoints (*p* = 0.032). Indeed, a high-risk was calculated for 22.0% at the first timepoint and 28.9% at the second.

Regarding the distributions of the BDI-II and PSS-10 scores across the independent variables, there were some significant relationships. Both the scores were differently distributed across gender, family cohesion, sexual orientation, economic situation, judgment about medical school choice, satisfying friendships with classmates, and climate among classmates (Appendix A). No significant correlation was found between age and BDI-II score (*p* = 0.633) or PSS-10 score (*p* = 0.200). 

The multivariable regression models are presented in Table 5. Participants who completed the survey at the second timepoint were more likely to score higher both on BDI-II and PSS-10. Moreover, being female, having poor family cohesion, negatively judging the medical school choice, and having unsatisfying friendships with classmates were positively associated with both the BDI-II and the PSS-10 scores. Instead, attending the sixth year were negatively associated with both the scores. In addition, a poor economic situation had a positive association with PSS-10 score.

## 4. Discussion

The present paper aimed to answer three main RQs. Study 1 (longitudinal sub-sample) explored whether medical students had higher levels of depressive symptoms and perceived stress during the pandemic compared with a pre-pandemic period (RQ1) and what variables were associated with these conditions during the pandemic adjusting for the baseline levels (RQ2). Then, Study 2 (repeated cross-sectional data) aimed to examine whether medical students had higher levels of these conditions during the pandemic compared with their peers during a pre-pandemic period (RQ3).

To answer RQ1, the results of Study 1 showed higher levels of depression and perceived stress during the pandemic in the longitudinal subsample. As already mentioned, other prospective studies conducted before and during the pandemic reported conflicting findings, showing increased, reduced, or unchanged symptoms [22,23,24,25]. These differences could be due to several factors, such as differences in year of training of participants, time frame before and during the pandemic, and country of the study, which may imply not only diverse sociocultural variables, but also different medical school training, COVID-19 epidemiological situation, and pandemic-related restrictive measures. Conversely, comparing the prevalence of depression and stress we found during the second timepoint with pre-pandemic reviews on medical students’ mental health, it is clear that our results revealed a higher presence of symptoms [1,7], thus confirming the findings reported by Mittal and colleagues [17].

It should be noted that studies covering the pre-clinical phase of medical school (i.e., the first three years, as in our longitudinal subsample) reported a rise in depressive symptoms over the three year period [37,38], suggesting that the adaptation to medical school may be challenging itself and thus we argue that this change that we saw might be partially due to the transition across medical school years. Moreover, other factors, and their changes in these three years, could be a great burden on students’ psychological health. Indeed, it is interesting to note that in the multilevel regression model of depressive symptoms the period of observation did not show a significant relationship with the outcome when adjusting for other variables. Specifically, in this model, family cohesion, sexual orientation, and judgment about the choice of medical school had a greater impact on depressive symptoms, confirming relationships with mental health found in previous studies [5,39,40,41]. On the other hand, the pandemic itself might have a role in potential changes of these relevant variables, e.g., changes in relationships with family [42,43] and in career choice perceptions [44,45,46]. Last, it should be highlighted that family cohesion and judgment about medical school choice had a significant association also in the multilevel model of perceived stress, thus highlighting their substantial importance that should be taken into account when planning interventions addressed to medical students’ wellbeing. 

The relevance of the judgment of medical school choice is highlighted also by the multivariable models executed to answer RQ2. Indeed, among the variables associated with depressive symptoms and stress during the pandemic adjusting for the baseline levels, a worsening of this judgment from the 1st to the 3rd year of medical school was associated with both the conditions, thus suggesting that the academic life strongly influences personal life and wellbeing. Adjustment to a new situation such as studying medicine might be a great psychological challenge and students would benefit from support in their adaptation to medical school [22]. In addition, certain pandemic-related variables were associated with the outcomes adjusting for the baseline levels. First, a perceived worsened psychological condition due to the pandemic had a significant relationship with both the outcomes, thus indicating that students may recognize that the pandemic is having a role in their poor mental health status. Then, the economic repercussions due to COVID-19 pandemic were shown to have a significant impact on stress, consistent with literature about determinants of mental health issues in youth [13,47]. Last, it should be noted that baseline levels of symptoms before the pandemic were associated with symptoms during the pandemic period of observation. This finding is consistent with a similar longitudinal study on medical students conducted before and during the pandemic [24]. In addition, the association between pre-existing mental health problems (potentially the baseline situation in the first months of the first year of medical school as in PRIMES) and mental health issues in medical students has been reported [48]. Moreover, COVID-19 could have a greater impact on individuals with mental health conditions compared with individuals without such conditions [49].

To solve the issue of the transition from 1st to 3rd year of medical school, RQ3 explored if students had higher levels of symptoms during the pandemic compared with their same-year peers during a pre-pandemic period. Our findings highlighted higher levels of depressive symptoms and stress during the pandemic, also adjusting for variables that might influence the outcomes [27,28], thus suggesting that students actually had a higher burden during the pandemic. We argue that these results give a meaningful insight of the actual impact of the pandemic on medical students’ mental health as the two timepoints of the repeated cross-sectional study are highly comparable since they possibly share the same sociocultural features and, in addition, many mental health-associated variables were recorded in both timepoints, thus giving further substantial evidence to the findings of the review conducted by Mittal and colleagues [17]. Furthermore, the other variables that resulted significant in the models of Study 2 were mainly the same that the ones commented above (i.e., family cohesion, sexual orientation, economic situation, judgment of medical school choice) and also female gender, consistently with depression epidemiology [50] and with data on medical students’ stress [5], and relationships with peers, as shown in other studies on medical students both before and during the pandemic [21,40]. In addition, attending the last year of medical school reduced the risk of symptoms. Although it has been reported a worsening of mental wellbeing across the first three pre-clinical years [37,38], this result seems to confirm the findings of a review showing a decreasing trend during the last years [3].

This study had some limitations that should be acknowledged. The main limitations are represented by the opportunistic sampling and the fact that no data about students who refused to participate were collected. Moreover, the study was performed in a single center, limiting its representativeness and generalizability. Last, in Study 2 the samples of the two timepoints were considered as independent samples, but it could be possible that some individuals who were at the 4th year in 2018 repeated the survey at the 6th year in 2020. However, this study has the strength to be one of the first work addressing medical students’ mental health in Italy using a prospective design, thus providing an infrequent and useful point of view.

## 5. Conclusions

This study showed that the levels of depressive symptoms and stress among medical students were substantially greater than before the pandemic, especially considering perceived stress, which was significantly higher in all the models adjusted for other relevant variables. In addition, during the pandemic, the most important variables associated with these two conditions were items related to this unprecedented situation, i.e., a perceived worsened psychological condition and economic repercussions due to the COVID-19, and the worsening of the judgment of the choice of medical school. Thus, with regard to the pandemic-related factors, universities should learn from these years and plan interventions (e.g., help for students with economic issues or bolstering university counselling services) to be implemented not only to face the current pandemic, but also to be effectively prepared for future emergency situations that may harm students’ health. Last, considering that the judgment of the career choice seemed to have a meaningful role in the mental health of students, the development of preventive and supportive interventions offered by high schools and, later, universities should certainly consider career planning as an essential component.

## Figures and Tables

**Table 1 jcm-11-05896-t001:** Descriptive analyses and McNemar analyses stratified by timepoint (Study 1).

Characteristic		Timepoint		*p*-Value
		First (*n* = 121) N (%)	Second (*n* = 121) N (%)	
Presence of depressive symptoms (BDI-II)	No	91 (79.1)	57 (49.6)	**<0.001**
	Yes	24 (20.9)	58 (50.4)	
Perceived stress categories (PSS-10)	Low stress	46 (39.7)	16 (13.8)	**<0.001**
	Moderate stress	61 (52.6)	66 (56.9)	
	High stress	9 (7.8)	34 (29.3)	
Off-site student	No	64 (53.8)	59 (49.6)	0.302
	Yes	55 (46.2)	60 (50.4)	
Living alone	No	113 (95.0)	114 (95.8)	1.000
	Yes	6 (5.0)	5 (4.2)	
Very poor/poor family cohesion	No	108 (89.3)	106 (87.6)	0.774
	Yes	13 (10.7)	15 (12.4)	
Relationship status	Single	67 (55.8)	62 (51.7)	0.458
	Involved	53 (44.2)	58 (48.3)	
LGBA sexual orientation	No	96 (81.4)	88 (74.6)	0.115
	Yes	22 (18.6)	30 (25.4)	
Family history of psychiatric disorders	No	88 (73.3)	92 (76.7)	0.481
	Yes	32 (26.7)	28 (23.3)	
Very poor/poor economic status	No	109 (90.8)	114 (95.0)	0.125
	Yes	11 (9.2)	6 (5.0)	
Negative judgment of medical school choice	No	108 (90.0)	90 (75.0)	**0.001**
	Yes	12 (10.0)	30 (25.0)	
Unsatisfying friendships with classmates	No	115 (99.1)	107 (92.2)	**0.021**
	Yes	1 (0.9)	9 (7.8)	
Climate among classmates judged as hostile/competitive	No	117 (97.5)	97 (80.8)	**<0.001**
	Yes	3 (2.5)	23 (19.2)	

Figures are expressed as frequencies and column percentages in brackets. *p*-value obtained via McNemar test or Marginal homogeneity test (significant *p*-values in bold). Abbreviations: BDI-II Beck Depression Inventory II; LGBA Lesbian Gay Bisexual Asexual; PSS-10 Perceived Stress Scale 10.

**Table 2 jcm-11-05896-t002:** Multivariable repeated-measures multilevel regression models (Study 1, Research Question 1).

Characteristic	BDI-II Score		PSS-10 Score	
	adjB (95% CI)	*p*-Value	adjB (95% CI)	*p*-Value
Age	1.06 (−0.41;2.53)	0.156	0.41 (−0.89;1.72)	0.533
Gender: female	−0.40 (−3.17;2.36)	0.775	0.92 (−1.49;3.34)	0.453
Second timepoint	3.28 (−0.13;6.69)	0.059	3.54 (0.54;6.53)	**0.021**
Very poor/poor family cohesion	5.51 (2.22;8.81)	**0.001**	3.01 (0.15;5.87)	**0.039**
LGBA sexual orientation	3.58 (1.04;6.11)	**0.006**	1.83 (−0.39;4.04)	0.106
Family history of psychiatric disorders	2.02 (−0.61;4.64)	0.132	2.47 (0.2;4.75)	**0.033**
Very poor/poor economic situation	−0.60 (−5.04;3.84)	0.791	0.34 (−3.47;4.15)	0.861
Negative judgment of medical school choice	7.44 (4.79;10.10)	**<0.001**	4.84 (2.59;7.1)	**<0.001**
Unsatisfying friendships with classmates	1.28 (−3.62;6.17)	0.609	1.85 (−2.48;6.19)	0.402
Climate among classmates judged as hostile/competitive	2.95 (−0.56;6.47)	0.100	2.43 (−0.43;5.29)	0.096
Living alone	2.25 (−2.60;7.09)	0.363	-	-
Off-site student	-	-	0.78 (−1.2;2.76)	0.440
Relationship status: involved	-	-	−0.36 (−2.24;1.52)	0.706

Significant *p*-values in bold. Abbreviations: adjB adjusted unstandardized coefficient; BDI-II Beck Depression Inventory II; CI Confidence Interval; LGBA Lesbian Gay Bisexual Asexual; PSS-10 Perceived Stress Scale 10.

**Table 3 jcm-11-05896-t003:** Multivariable linear regression models (Study 1, Research Question 2).

Characteristic	BDI-II Score		PSS-10 Score	
	adjB (95% CI)	*p*-Value	adjB (95% CI)	*p*-Value
Age	0.6 (−1;2.2)	0.457	1.18 (−0.01;2.37)	0.051
Gender: female	−2.11 (−5.14;0.92)	0.170	−0.34 (−2.61;1.92)	0.764
BDI-II score during the first timepoint	0.55 (0.36;0.75)	**<0.001**	−	−
PSS-10 BDI-II score during the first timepoint	−	−	0.54 (0.39;0.69)	**<0.001**
Feeling of loneliness during the pandemic:				
No	Ref.		−	−
Yes, more than usual	−1.2 (−4.56;2.16)	0.480	−	−
Yes, as usual	2.58 (−2.48;7.64)	0.314	−	−
Existential reflections related to COVID-19:				
No	Ref.		−	−
Yes, positively stimulating	0.27 (−3.52;4.05)	0.890	−	−
Yes, deeply distressing	3.05 (−0.76;6.86)	0.115	−	−
Overall psychological condition changed due to COVID-19:				
No	Ref.		Ref.	
Yes, improved	−5.44 (−14.45;3.58)	0.234	−1.26 (−8.08;5.56)	0.714
Yes, worsened	5.75 (2.52;8.98)	**0.001**	2.84 (0.52;5.16)	**0.017**
Economic repercussions due to COVID-19	−	−	3.32 (0.61;6.02)	**0.017**
Very poor/poor economic situation	−	−	2.22 (−1.98;6.42)	0.296
Judgment on medical school choice:				
Unchanged	Ref.		Ref.	
Worsened	9.22 (5.58;12.86)	**<0.001**	5 (2.36;7.63)	**<0.001**
Improved	−5.68 (−12.38;1.01)	0.095	−7.11 (−11.74;−2.48)	**0.003**
Economic situation:				
Unchanged	Ref.		−	−
Worsened	5.68 (−0.5;11.85)	0.071	−	−
Improved	1.6 (−14.63;17.83)	0.846	−	−
Family cohesion:				
Unchanged	−	−	Ref.	
Worsened	−	−	4.78 (−0.35;9.91)	0.067
Improved	−	−	3.47 (−2.27;9.21)	0.233

Significant *p*-values in bold. Abbreviations: adjB adjusted unstandardized coefficient; BDI-II Beck Depression Inventory II; CI Confidence Interval; LGBA Lesbian Gay Bisexual Asexual; PSS-10 Perceived Stress Scale 10.

**Table 4 jcm-11-05896-t004:** Descriptive analyses and chi-squared analyses stratified by timepoint (Study 2).

Characteristic		Overall Sample (*n* = 705)	Timepoint		
		N (%)	First (*n* = 283) N (%)	Second (*n* = 422) N (%)	*p*-Value
Presence of depressive symptoms (BDI-II)	No	402 (57.7)	177 (64.1)	225 (53.4)	**0.005**
	Yes	295 (42.3%)	99 (35.9)	196 (46.6)	
Perceived stress categories (PSS-10)	Low stress	138 (19.7)	66 (23.8)	72 (17.1)	**0.032**
	Moderate stress	378 (54.1)	150 (54.2)	228 (54.0)	
	High stress	183 (26.2)	61 (22.0)	122 (28.9)	
Age		24 (22–24)	23 (22–24)	24 (23–24)	**0.036**
Gender	Male	230 (32.8)	90 (32.0)	140 (33.3)	0.735
	Female	472 (67.2)	191 (68.0)	281 (66.7)	
Year of course	4th	344 (48.8)	143 (50.5)	201 (47.6)	0.450
	6th	361 (51.2)	140 (49.5)	221 (52.4)	
Off-site student	No	296 (42)	118 (41.8)	178 (42.2)	0.929
	Yes	408 (58)	164 (58.2)	244 (57.8)	
Living alone	No	659 (93.5)	255 (90.1)	404 (95.7)	**0.003**
	Yes	46 (6.5)	28 (9.9)	18 (4.3)	
Very poor/poor family cohesion	No	596 (84.7)	235 (83.3)	361 (85.5)	0.425
	Yes	108 (15.3)	47 (16.7)	61 (14.5)	
Relationship status	Single	276 (39.1)	119 (42)	157 (37.2)	0.196
	Involved	429 (60.9)	164 (58)	265 (62.8)	
LGBA sexual orientation	No	527 (75.4)	225 (80.9)	302 (71.7)	**0.006**
	Yes	172 (24.6)	53 (19.1)	119 (28.3)	
Family history of psychiatric disorders	No	522 (74.3)	211 (75.1)	311 (73.7)	0.679
	Yes	181 (25.7)	70 (24.9)	111 (26.3)	
Very poor/poor economic status	No	661 (93.9)	263 (93.3)	398 (94.3)	0.568
	Yes	43 (6.1)	19 (6.7)	24 (5.7)	
Negative judgment of medical school choice	No	468 (66.5)	181 (64.2)	287 (68)	0.292
	Yes	236 (33.5)	101 (35.8)	135 (32)	
Unsatisfying friendships with classmates	No	615 (89.4)	246 (88.8)	369 (89.8)	0.685
	Yes	73 (10.6)	31 (11.2)	42 (10.2)	
Climate among classmates judged as hostile/competitive	No	559 (79.6)	223 (79.6)	336 (79.6)	0.994
	Yes	143 (20.4)	57 (20.4)	86 (20.4)	

Figures are expressed as frequencies and column percentages in brackets. *p*-value obtained via chi-squared test. (Except for age: expressed as median and interquartile range in brackets; *p*-value obtained via Mann-Whitney U test) (significant *p*-values in bold). Abbreviations: BDI-II Beck Depression Inventory II; LGBA Lesbian Gay Bisexual Asexual; PSS-10 Perceived Stress Scale 10.

**Table 5 jcm-11-05896-t005:** Multivariable linear regression models (Study 2, Research Question 3).

Characteristic	BDI-II Score		PSS-10 Score	
	adjB (95% CI)	*p*-Value	adjB (95% CI)	*p*-Value
Age	0.33 (−0.24;0.9)	0.257	0.39 (−0.05;0.84)	0.082
Gender: female	3.54 (2.12;4.97)	**<0.001**	4.17 (3.04;5.29)	**<0.001**
Second timepoint	3.27 (1.88;4.65)	**<0.001**	2.19 (1.12;3.27)	**<0.001**
Sixth year of course	−2.89 (−4.61;−1.17)	**0.001**	−1.85 (−3.19;−0.51)	**0.007**
Very poor/poor family cohesion	3.55 (1.62;5.48)	**<0.001**	2.3 (0.78;3.81)	**0.003**
LGBA sexual orientation	0.57 (−1.01;2.15)	0.478	0.95 (−0.28;2.19)	0.130
Family history of psychiatric disorders	0.79 (−0.74;2.32)	0.312	0.14 (−1.06;1.33)	0.820
Very poor/poor economic situation	2.64 (−0.15;5.44)	0.064	3.24 (1.06;5.42)	**0.004**
Negative judgment of medical school choice	7.64 (6.17;9.1)	**<0.001**	4.9 (3.76;6.03)	**<0.001**
Unsatisfying friendships with classmates	4.68 (2.43;6.92)	**<0.001**	1.89 (0.14;3.64)	**0.034**
Climate among classmates judged as hostile/competitive	1.28 (−0.47;3.03)	0.151	0.22 (−1.14;1.58)	0.752
Living alone	1.51 (−1.19;4.2)	0.273	-	-
Off-site student	-	-	0.33 (−0.73;1.38)	0.544
Relationship status: involved	-	-	−0.54 (−1.63;0.55)	0.332

Significant *p*-values in bold. Abbreviations: adjB adjusted unstandardized coefficient; BDI-II Beck Depression Inventory II; CI Confidence Interval; LGBA Lesbian Gay Bisexual Asexual; PSS-10 Perceived Stress Scale 10.

## Data Availability

All relevant data are within the paper. The dataset is available from the corresponding author on reasonable request.

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
