# Peer review of "The Consequences of the Pandemic on Medical Students’ Depressive Symptoms and Perceived Stress: A Repeated Cross-Sectional Survey with a Nested Longitudinal Subsample"

_jcm, 2022, doi:10.3390/jcm11195896_

Round 1

Reviewer 1 Report

Dear authors,

Thank you for the opportunity to review your paper.

The manuscript is well written and follow a rigorous mythological design. However, a few issues should be considered in the revised version:

Introduction

·       I suggest that research questions to be numbered just as RQ1, ... RQ3, without a, b ...

·       In the end of the introduction, please summarize in a paragraph the content of the remaining sections.

Line 90, 96, 106 ...: University of Torino or University of Turin?

I suggest that the paper include both Discussion and Conclusion sections. Moreover, the theoretical and practical implications should be outlined.

Author Response

We would like to thank the reviewer for these useful comments. Our response is in attachment

Reviewer 2 Report

Very interesting topic, the researcher did a great job to explore the pandemic impact on medical students’ mental health in Italy. However, the article could be strengthened through:

Line 59: Introduce every acronym before using it in the text. The first time you use the term, put the acronym in parentheses after the full term, like coronavirus disease of 2019.

Line 81: (a) I think you mean (c).

Add more towards scope of the problem in introduction section.

Have you obtained permission to use the tools?

PRIMES Validity and reliability needs to be addressed.

Line 151: the researchers mentioned that they excluded some variables from PRIMES; have you modified the tool? If so, have you obtained the permission to do so? Also, have you tested the validity and the reliability after modifying the tool? Have you conducted a plot study?

Good luck

Author Response

(The authors gave the same response as above.)
